# Clinical Decision Support Systems for Diagnosis in Primary Care: A Scoping Review

**DOI:** 10.3390/ijerph18168435

**Published:** 2021-08-10

**Authors:** Taku Harada, Taiju Miyagami, Kotaro Kunitomo, Taro Shimizu

**Affiliations:** 1Department of General Medicine, Showa University Koto Toyosu Hospital, Tokyo 135-8577, Japan; hrdtaku@gmail.com; 2Department of Diagnostic and Generalist Medicine, Dokkyo Medical University Hospital, Tochigi 321-0297, Japan; 3Department of General Medicine, Juntendo University Faculty of Medicine, Tokyo 113-8421, Japan; tmiyaga@juntendo.ac.jp; 4Department of General Medicine, Kumamoto Medical Center, Kumamoto 860-0008, Japan; m05035kk@yahoo.ac.jp

**Keywords:** clinical decision support systems, diagnostic accuracy, health information technology, primary care

## Abstract

Diagnosis is one of the crucial tasks performed by primary care physicians; however, primary care is at high risk of diagnostic errors due to the characteristics and uncertainties associated with the field. Prevention of diagnostic errors in primary care requires urgent action, and one of the possible methods is the use of health information technology. Its modes such as clinical decision support systems (CDSS) have been demonstrated to improve the quality of care in a variety of medical settings, including hospitals and primary care centers, though its usefulness in the diagnostic domain is still unknown. We conducted a scoping review to confirm the usefulness of the CDSS in the diagnostic domain in primary care and to identify areas that need to be explored. Search terms were chosen to cover the three dimensions of interest: decision support systems, diagnosis, and primary care. A total of 26 studies were included in the review. As a result, we found that the CDSS and reminder tools have significant effects on screening for common chronic diseases; however, the CDSS has not yet been fully validated for the diagnosis of acute and uncommon chronic diseases. Moreover, there were few studies involving non-physicians.

## 1. Introduction

Diagnosis by primary care physicians (PCPs) is an important task; however, there is always a risk of diagnostic error in the task of diagnosis. Diagnostic errors are the greatest threat to patient safety in primary care [1]. A recent study estimated that approximately 5% of adult patients in the United States experience diagnostic errors in outpatient settings every year [2]. Estimates from diagnostic error rates in selected research studies indicate that 12 million Americans suffer from diagnostic errors in primary care alone each year [2,3]. The same study found that 33% of these diagnostic errors led to “serious permanent injury” or “immediate or inevitable death”. This translates into at least 4 million people seriously harmed, including at least 1.7 million people who died, due to diagnostic errors [3]. Thus, the prevention of diagnostic errors in primary care is an urgent issue. The World Health Organization (WHO) recently noted the importance of safety and diagnostic accuracy in primary care [4], and diagnostic issues that harm patients through errors or delays in testing and treatment have emerged as a matter of grave concern in global safety [5].

Primary care commonly involves a large number of patients and decision-making in the face of uncertainty [6]. In addition, the characteristics of primary care (first contact, accessible, continuous, comprehensive, and coordinated care) make it an area at high risk of diagnostic errors [7]. More than half of medical malpractice cases against general practitioners are due to diagnostic errors [8], and their prevention requires prompt action. In primary care, where the risk of diagnostic errors is high, there are high expectations for improving the diagnostic process through CDSS and health IT technologies [5].

One of the health IT products is the clinical decision support system (CDSS). CDSS is not intended to replace a clinician’s assessment, but rather to facilitate the clinician’s correct assessment and reasoning through suggestions and alerts [1,9]. CDSS performs various functions, including giving reminders, alerting users of prescription interactions and test results, interpreting tests, predicting mortality based on epidemiological data, assisting in diagnosis, and calculating drug doses [10].

CDSS has been demonstrated to enhance the quality of care in a variety of healthcare settings, including hospitals and primary care centers [1,11]. In 2007, the US government published “A Roadmap for National Action on Clinical Decision Support” to encourage the introduction of CDSS in electronic health records (EHRs) [12]. In 2013, 41% of US hospitals with EHRs were estimated to have CDSS, and in 2017, 40.2% had advanced CDS capabilities [13]. In the UK, the government introduced Isabel software as a national healthcare system [14], and an artificial intelligence (AI)-powered triage and diagnosis system called Babylon is in operation, although some barriers remain [15]. The CDSS market is currently dominated by North America and is estimated to reach USD 1.33 billion by 2021 and USD 2.24 billion by 2026 [16]. Conversely, language remains a barrier for introduction in non-English speaking countries [17]. In developing countries, barriers have been pointed out for EHR implementation before CDSS [18].

While CDSS is useful for improving medical care in the primary care setting, few studies have been conducted in the diagnostic setting [1], and there are few reports showing measurable clinical effects [19]. Additionally, CDSS use has not been promoted due to physicians’ negative perceptions and prejudices toward CDSS as a diagnostic aid, complex data entry, and gaps in data use [13]. Moreover, there are barriers to widespread CDSS use and shortcomings of CDSS: CDSS disrupts the workflow of healthcare professionals, increases the time required to complete tasks, increases cognitive load, and decreases time spent with patients (especially in standalone systems). CDSS has several disadvantages, including overriding by physicians, alert fatigue, and risk of deskilling of healthcare providers due to long-term use [13]. In primary care, concerns about reliability, impact on workflow, and incompatibility between the gatekeeper role and CDSS recommendations are considered barriers to adoption [20]. One study found that solo practices had significantly lower rates of CDSS use, regardless of EHR use [21]. Another problem is that cost-effectiveness is unknown [13]. To eliminate barriers to CDSS adoption, various factors such as internal and external environments, individuals, and interventions have been pointed out [22]. Particularly, to successfully eliminate barriers, user-centered design and analysis of impact on performance improvement are needed [23,24].

We conducted a scoping review to verify the usefulness of CDSS in the diagnostic domain in primary care and to identify research gaps and areas of uncertainty that require further exploration.

## 2. Materials and Methods

This scoping review was based on the methodology of the Preferred Reporting Items for Systematic Reviews and Meta-Analyses Extension for Scoping Reviews (PRISMA-ScR) statement [25]. The primary review questions were: (1) what is the current status of the usefulness of the CDSS in the diagnostic area of primry care? and (2) can we identify research gaps and areas of uncertainty from existing knowledge?

### 2.1. Search Strategy

We searched PubMed/MEDLINE for articles written in English and published until March 2020. Suitable search terms were chosen to cover the three dimensions of interest: decision support systems, diagnosis, and primary care. The query used the Medical Subject Headings (MeSH) terms, “Decision Support Systems, Clinical” [MeSH] and “Primary Health Care” [MeSH].

Two of the authors (TM and TH) independently screened the titles and abstracts of the retrieved articles to assess their relevance based on the eligibility criteria. A third author (KK) made the final decision on papers regarding which there was no consensus between TM and TH. Backward citation tracking was also performed to identify additional relevant articles. Finally, full-text versions of the articles that were found relevant by the two reviewers were reviewed.

### 2.2. Eligibility Criteria

The following inclusion criteria were used for the selection of relevant studies: (1) the study should have evaluated the implementation of a CDSS in primary care and (2) CDSS included everything from simple reminders to complex computer clinical decision support systems, all of which are relevant to the diagnostic process. Additionally, in this study, reminder tools were also included in the CDSS. Studies were excluded if they were (1) related to treatment or management; (2) targeted the accessibility of CDSS in primary care settings; (3) not real clinical studies using simulations or scenario cases; (4) studies on CDSS for caregivers; (5) studies in the pediatric population; and (6) review articles and protocol studies.

## 3. Results

### 3.1. General Overview

The search query returned 580 articles and backward citation tracking included 22 articles. After analyzing the titles and abstracts, 551 articles that were not relevant to the research question were discarded. Finally, 51 full-text articles were reviewed, among which 25 articles that did not satisfy the eligibility criteria were excluded. The corresponding PRISMA flow diagram is shown in Figure 1.

Twenty-six clinical studies (17 randomized control trials [RCTs] and nine non-RCTs) were included (Table 1). They comprised 10 studies on combined outcomes including malignancy prevention, vaccine uptake, and lifestyle prevention [26,27,28,29,30,31,32,33,34,35]; three studies on malignancy (one each on breast cancer, cervical cancer, and colorectal cancer) [36,37,38]; five studies on cardiovascular risk factors (one each on diabetes mellitus [DM], dyslipidemia [DLP], obesity, abdominal aortic aneurysm [AAA], and chronic kidney disease [CKD]) [39,40,41,42,43]; two studies on musculoskeletal diseases (one each on osteoporosis and function) [44,45]; three studies on infectious diseases (two on hepatitis B virus [HBV] and one on human immunodeficiency virus [HIV]) [46,47,48]; and others included one study on depression, one study on dementia, and one study on domestic violence [49,50,51]. Common chronic diseases were the main target, and none of the studies included common acute diseases or uncommon chronic diseases. In almost all studies, CDSS had significant results in the screening and diagnosis of chronic diseases.

### 3.2. Effect of CDSS on the Screening and Diagnosis of Composite Outcomes

The effect of the recommended multiple screening reminder as a composite outcome was assessed in five RCTs [26,27,28,29,30], all of which showed significantly positive results.

Two studies evaluated the impact of CDSS on clinical diagnosis without focusing on specific diseases or multiple outcomes [34,35]. Vetter’s study on nurse practitioners (NPs) in a home-based primary care setting showed that the introduction of the CDSS improved the diagnostic accuracy and appropriate documentation [34]. A retrospective study by Shimizu et al. [35] in an outpatient department of a community-based hospital examined the effects of CDSS and found that it significantly reduced diagnostic errors (odd’s ratio [OR] 15.21).

### 3.3. Effect of CDSS on the Screening and Diagnosis of Cancer

Hamilton et al. [31] investigated whether adding risk assessment tools would improve testing and diagnosis of lung and colorectal cancer in a before-and-after cohort study involving 165 clinics and 614 primary care physicians, using a nested qualitative study. They found 2593 uses of risk assessment tools with increased correct testing and diagnosis of lung and colorectal cancers after the addition.

Murphy, in an RCT, investigated whether E-triggers, based on electronic medical records, could reduce the time to diagnosis of colorectal, lung, and prostate cancers in two primary care practices, with 72 primary care physicians divided equally into intervention and target groups. Of the 10,673 patients, E-trigger was triggered in 1256 patients, resulting in a significant decrease in the time to diagnostic evaluation for colorectal and prostate cancers; however, no significant difference was observed for lung cancer [32].

Price et al. [33] conducted a cross-sectional study in primary care clinics to determine the relationship between the use of cancer decision support tools and the number of referrals for suspected malignancy (2-week wait). The results showed that there was no significant difference between the use of cancer decision support tools and the number of referrals, suggesting the possibility of the underuse of CDSS in primary care in the United Kingdom.

Three studies were applicable to CDSS interventions for a single malignancy. Burack and Gimotty [36] conducted an RCT in three primary care practices to evaluate the sustained effect of computerized reminders in the second year of the intervention. The results showed that over the 2 years, the reminder group had significantly higher mammography uptake than in the pre-intervention period. Burack et al. [37], in an RCT of 5801 women, evaluated the influence of reminders to patients and/or physicians on PAP smear practices and found no significant effect. Sequist et al. [38] conducted an RCT of reminders to 21,860 patients and 110 primary care physicians in 11 ambulatory care centers to determine the impact of reminders on colorectal cancer screening. The screening rate was similar and did not significantly increase between patients of physicians who received email reminders and the control group (41.9% vs. 40.2%, *p* = 0.47).

### 3.4. Effect of CDSS on the Screening and Diagnosis of Cardiovascular Risk Factors

Five CDSS studies were applicable for screening and diagnosing cardiovascular risk factors. They included three RCTs, one non-RCT, and one retrospective observational study. The participants were screened for CKD, AAA, obesity, DM, and DLP. Litvin et al. [39] evaluated the effect of implementing the CDSS for the identification and management of CKD in 11 primary care clinics. The results showed significant improvement in the screening and monitoring of albuminuria over 2 years. Lee et al. [40] measured the effect of CDSS in 1874 clinical sites over an 8-month period among NPs of two specialties (acute care and family) and showed that the CDSS group had 11.3% more diagnoses and 37% fewer false negatives than the control group. The introduction of CDSS has improved the diagnosis of obesity. In a before-after retrospective observational study, Chaudhry et al. [41] investigated the screening construction rate of AAA before and after the introduction of CDSS in male patients aged 65–75 years who visited the clinic in 2007 and 2008. The overall screening rate increased significantly (13%), and the percentage of patients in the completed-screening group improved by approximately fivefold, from 3.2% to 18.2%.

Kenealy et al. [42] conducted an RCT of an intervention that included computer-based reminders to improve screening for DM. The duration of the study was 2 months, and 107 family physicians participated. The results showed that computer-based reminders significantly improved the screening rate for DM (OR, 1.49). Wyk et al. [43] conducted an RCT of 38 clinics, 77 physicians, and 87886 patients in the Netherlands to determine the effect of alerts on improving dyslipidemia screening. Each clinic was assigned to one of three groups: receive alerts, on-demand support, or no intervention. After 12 months of follow-up, screening occurred 65% of the time in the alert group, compared to 35% in the on-demand group, and 25% in the target group, with a significantly higher screening rate in the alert group. The frequency of treatment for patients needing treatment was also significantly higher in the alert group (66% vs. 40% vs. 26%).

### 3.5. Effect of CDSS on the Screening and Diagnosis of Musculoskeletal Conditions

Rubenstein et al. [44] conducted an RCT to determine the impact of implementing the CDSS, which provides screening and feedback for improving physical functioning in older adults. Seventy-three internists and 557 patients in a primary care clinic participated; patients in the CDSS group had significantly less frequent functional decline and significantly improved emotional well-being scores compared with the control group.

DeJesus et al. [45] conducted a before-and-after retrospective observational study to measure the effect of CDSS on osteoporosis screening. Eligible patients were women aged ≥ 65 years who had never undergone a bone mineral density test and were seen in a primary care clinic. They found that the overall screening rate improved significantly from 80.1% to 84.1%, and completion of screening after the visit increased from 5.87% to 9.79%, an improvement of 66.7%.

### 3.6. Effect of CDSS on the Screening and Diagnosis of Infectious Diseases

DeSilva et al. [46] conducted a pilot study at nine clinics to screen for HBV infection in people born outside the United States. Eligible patients were aged ≥ 12 years and from countries with HBV infection rates of 2% or higher. E-alerts were triggered for more than 4500 patients between July 2012 and March 2013, and in 14.0% of patients, healthcare providers responded to the trigger; six previously unrecognized HBV-infected patients were identified. Although the usefulness of the triggers was demonstrated, there was no significant difference between passive and active interventions, and the response rate to the triggers decreased yearly.

Chak et al. [47] conducted a study in the United States to measure the effect of E-alerts in screening foreign-born HBV high-risk populations. Over a period of 1 year, 2987 patients were included in the study and the intervention group had significantly more screening tests performed than the control group (OR 2.64), demonstrating the usefulness of E-alerts in HBV screening.

An RCT was conducted by Sundaram et al. [48] to determine the efficacy of a computerized reminder and feedback intervention in improving HIV screening among 32 physicians in five primary care clinics. The results showed that the intervention did not significantly improve HIV screening, possibly due to the overall low rate of HIV screening and barriers to reminders.

### 3.7. Effect of CDSS on the Screening and Diagnosis of Other Diseases

Miller et al. [49] investigated whether drug-related E-triggers improve depression screening in a 3-year cross-sectional descriptive study. Primary care physicians screened for depression in 2.1% of patients, with a significant increase in screening rates, especially when moderate or high warnings were given.

Downs et al. [50] conducted an unblinded, clustered, randomized, controlled study to determine whether CDSS improves the detection of dementia in primary care. Thirty-six clinics were randomly assigned to the control, workshops, tutorials on CD-ROM, and CDSS embedded in the electronic health record groups. The CDSS significantly improved the detection of dementia compared to the control group in 450 records (30% vs. 11%).

An RCT of 11 family physicians and 282 patients was conducted by Ahmad et al. [51] to evaluate whether computer-assisted screening by family physicians could improve the detection of women at risk for intimate partner violence and control (IPVC); they found that CDSS significantly improved the detection of IPVC compared to the control group (18% vs. 9%).

## 4. Discussion

The results of this scoping review show that the CDSS and reminder tools have significant results in screening for common chronic diseases; however, the CDSS has not yet been fully validated for the diagnosis of acute disease and uncommon chronic diseases.

The clinical usefulness of CDSS in acute illness and uncommon chronic diseases seems promising. A previous study showed a 17% improvement in the diagnosis of acute abdominal pain when CDSS was introduced in an emergency room [52]. Moreover, though simulation-based, the usefulness of CDSS for primary care physicians in various diseases such as orthopedic diseases [9], ophthalmic diseases [53], and skin malignancies [54] has been suggested. Farmer developed a knowledge-based CDSS to aid in the diagnosis of shoulder disorders in the primary care setting based on computer science literature and orthopedic opinion. Although accuracy varied with each shoulder disease, the CDSS diagnostic results for 93 case studies had a sensitivity of 91%, specificity of 98%, positive likelihood ratio of 53.12, and a negative likelihood ratio of 0.08 [9]. López et al. conducted a study on the functionality and reliability of OphthalDSS, a mobile application for the diagnosis of anterior segment ocular disease diagnosis. Fifty primary care physicians in Spain used OphthalDSS and evaluated the results; 70% of physicians were satisfied with the functionality and 95% of physicians rated it as reliable [53]. Gerbert et al. investigated whether the CDSS could support primary care physicians in triaging lesions suggestive of cutaneous malignancies (basal cell carcinoma and squamous cell carcinoma). In a study of 20 primary care physicians presenting 15 skin lesions and comparing triage options, the percentage of incorrect triage choices decreased from 36.7% to 13.3% when CDSS was used [54]. Although the above study was not conducted in a clinical setting, the results show that the CDSS can be expected to improve diagnosis for a variety of diseases in primary care.

Rapid diagnosis of rare diseases by primary care physicians remains a major challenge [55]. The UK Strategy for Rare Diseases advocates the use of ‘effective IT support’ for the diagnosis of infrequent collagen diseases in primary care. [55]. The usefulness of CDSS for rare diseases and difficult-to-diagnose cases has been previously shown on a simulation-based research [56,57]. In addition, Ronicke et al. investigated the effect of a CDSS called Adax Dx in the setting of an outpatient clinic for rare inflammatory systemic diseases [58]. A retrospective analysis of the diagnostic process in 93 confirmed diagnoses (84 cases of collagen disease) showed that 53.8% of cases had accurate diagnosis as one of the top five differential diagnoses earlier than physicians’ clinical diagnosis; in 37.6% of cases, positive diagnosis was the top differential diagnosis, suggesting the usefulness of CDSS for rare diseases. In a study by Rees et al., which developed and validated a risk prediction model to support early diagnosis of systemic lupus erythematosus (SLE), the sensitivity was 34% and the specificity was 90%; thus, early diagnosis is not possible in about two-thirds of cases. In addition, the absolute risk prediction value for SLE is usually less than 1% due to the low frequency of SLE [59]. Pearce also used primary care data to look for signs that could help in the early diagnosis of granulomatosis with polyangiitis but reported no useful signs for early detection [60]. These results suggest that there are two problems that need to be solved before introducing CDSS for the diagnosis of rare diseases in primary care settings [55]. First, it is difficult to build a model with high specificity for some diseases, and second, the lower the frequency of the disease, i.e., the lower the pre-test probability, the lower the positive predictive value. The best way to solve these problems is to improve the pre-test probability, including the exclusion of common diseases, and enhance the clinical reasoning by primary care physicians. In order to diagnose rare diseases in primary care without delay, how to use the CDSS in the augmentation of routine care will be one of the factors that primary care physicians will need to consider in the future.

Although this study demonstrated the utility of the CDSS in the screening and diagnosis of malignancies in a primary care setting, several limitations remain. The barriers to CDSS implementation and cost-effectiveness, as well as the long-term prognosis of early screening with CDSS and the benefits of CDSS diagnosis and transcription for symptomatic patients, are not yet known [61].

In our scoping review, two studies on NPs were included. In recent years, there has been a paradigm shift in the diagnostic process, with diagnoses no longer being made by physicians alone, but rather as a collaborative process involving patients and multiple professionals [62]. In a scoping review conducted by Abdellatif et al. [63] on the benefits of CDSS in nursing home settings, not only physicians but also nurses and pharmacists were found to benefit from CDSS in correcting malnutrition, pressure ulcer prevention, drug prescription, and disease management. Therefore, it is extremely important to examine the usefulness of CDSS for medical professionals other than physicians and how to utilize CDSS in multidisciplinary teams. The management of CDSS in primary care by all types of healthcare professionals and multidisciplinary teams is an important theme for the future.

One of the issues to be addressed in the future is how to adapt the CDSS to primary care settings. According to a systematic meta-review conducted by Nurek et al. [1], there are three main challenges in integrating CDSS in the field: (1) a more standardized computable approach to knowledge representation, (2) one that can be readily updated as new knowledge is gained, and (3) the need for it to trigger at appropriate points in cognitive workflow. Moreover, it also poses the barrier of failure to use dynamic vocabulary tools and to integrate with electronic health records. In fact, of the 26 articles reviewed in the present study, 20 showed an association with factors related to the use of CDSS and its barriers, including time-varying barriers, workflow, situational factors, limited clinic time, warning fatigue, and healthcare provider factors [30,31,33,34,36,37,38,39,41,42,43,44,48,49,50,51]. Patient-centered medicine is a key point in the field of primary care. When digital technologies are used in primary care, a triangle of patient-eHealth-PCPs is created, implying that the direct interaction between patients and PCPs may be affected by eHealth, and the role, importance, and meaning of human interaction in primary care must be reconsidered [64]. There is a viewpoint that digital health can enhance shared decision-making (SDM); however, others advocate that attention should be paid to maintaining “humanity” [64]. In the area of diagnosis, problems with medical information technology include that it cannot weigh in order of importance, cannot deal with comorbidities, cannot consider time series or the context of the patient, and cannot obtain input information by itself [65]. However, these disadvantages are not compatible with patient-centered medicine. It is necessary to carefully consider how to augment the advantages and disadvantages of patient-centered medicine and shared decision-making, and how to integrate them into face-to-face care without compromising quality or safety [66]. In addition, in CDSS, it is common for all input information to be entered into the system before the diagnostic process; in contrast, it is believed that in humans, the concurrent process of history taking and diagnostic thinking can easily lead to a more accurate diagnosis by calibrating the clinical information and diagnosis accordingly [67]. Moreover, there are concerns at present about the validity of the quality of medical history taking by artificial intelligence (AI) [68]; thus, the role of medical history taking in medicine practice is critical. Given that fact face-to-face communication with patients is the predominant mode of diagnosis in primary care, one of the issues to be solved is how to adapt CDSS into the current field of medicine from the user’s perspective, and how to turn its disadvantages into opportunities. This is an issue that needs to be resolved in the field of primary care medicine.

The last issue that should be added is the demand for human resources involved in the development of such diagnostic AI. In particular, the development of diagnostic AI requires the presence of clinical diagnosticians who can advise on its validation and design. Therefore, it will be important to train clinicians, including diagnosticians, who are interested in diagnostic AI and have an understanding of medical informatics.

### Strengths and Limitations

In our opinion, this is the first review in the literature to focus on the category of “diagnosis” in primary care settings. In this study, we found that the CDSS is mainly used for the diagnosis and screening of common chronic diseases, and that future directions include the diagnosis of acute and rare diseases and the use of CDSS by non-physician healthcare providers.

However, this study has three limitations. First, we did not evaluate the quality of each of the included study. Second, we did not examine the barriers to the introduction of CDSS in clinical practice. Third, because it is a scoping review, there are limitations related to publication bias. In particular, some studies with negative results may not have been published.

## 5. Conclusions

We conducted a scoping review to determine the usefulness of the CDSS diagnosis in primary care and to identify areas that need to be explored. We found that the CDSS and reminder tools have significant results in screening for common chronic diseases; however, the CDSS has not yet been fully validated for the diagnosis of acute and uncommon chronic diseases. There have also been a few studies involving non-physicians.

Future research on the diagnosis of CDSS in primary care should focus on the usefulness of CDSS for acute and infrequent diseases, its use in situations involving patients and non-physicians, and its diagnostic accuracy. Fortunately, the issues that need to be addressed in each disease group for each clinical setting have themselves been identified. Therefore, with the progress of future research on CDSS, these issues will be solved separately.

At the same time, the value and role of the CDSS will change depending on how medical professionals in the field think about how to use the CDSS, reflecting the clinical setting and situation. In this era of AI and human augmentation, CDSS, especially in the field of diagnosis, is still in its infancy. However, the help of CDSS, a relatively new AI-based technology, will result in improved diagnoses, which is in line with the major goal of the current entire medical community. As such, the potential growth rate of this field in the future will contribute greatly to improving the quality of healthcare.

## Figures and Tables

**Figure 1 ijerph-18-08435-f001:**
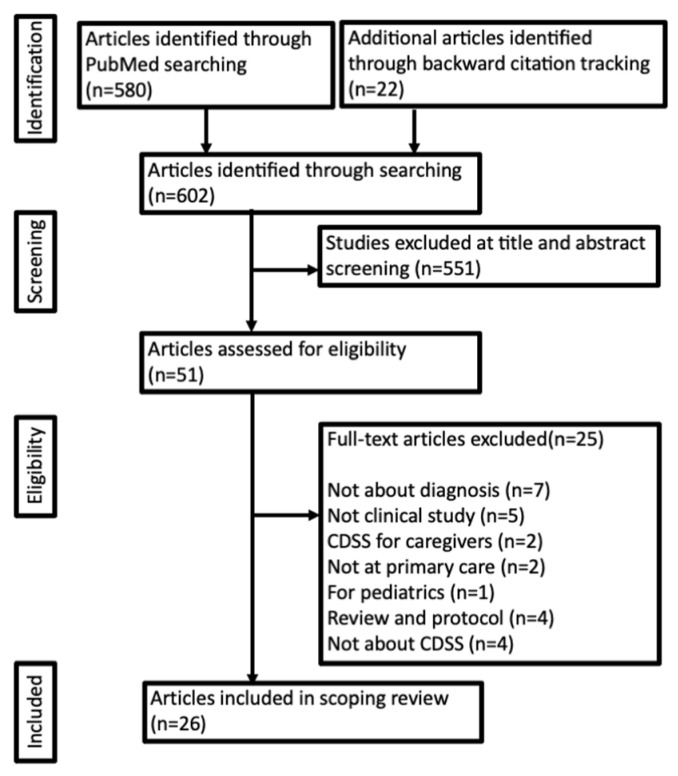
PRISMA flow diagram.

**Table 1 ijerph-18-08435-t001:** Characteristics of the 26 clinical studies included in this scoping review.

Reference	Participants	Age	Study Design	Target Disease	Process Outcome	Main Outcome	Main Observation	Significant Difference	Evaluator	Support Method
Rosser et al. 1991, Canada [26]	8502 patients	Over 15 years old	Randomized controlled trials	combined outcomes	Screening rate	Rates of completion of the preventive procedures	Five screening procedures (Flu vaccine, measure blood pressure, Assess smoking status, Papanicolaou smear, Tetanus vaccine)	○	Physician	Reminder
Ornstein et al. 1991, USA [27]	7397 patients	40.0 ± 17.3	Randomized controlled trials	combined outcomes	Screening rate	Adherence to preventive serivices	Five recommended preventive services (cholesterol measurements, fecal occult blood testing, mammography, Papanicolaou smears, and tetanus vaccine)	○	Physician	Reminder
McPhee et al. 1991, USA [28]	2331 encount	No details available	Randomized controlled trials	combined outcomes	Screening rate	Rates of completion of the preventive procedures	Nine cancer prevention services and screenings (stool occult-blood test, rectal examination, pelvic examination, Papanicolaou’s smear, breast examination, smoking assessment, smoking counseling, dietary assessment, dietary counseling, sigmoidoscopy, mammography)	○	Physician	Reminder
McPhee et al. 1989, USA [29]	1936 records	Varies by criteria	Randomized controlled trials	combined outcomes	Screening rate	Rates of completion of the cancer screening procedures	Seven cancer prevention screenings (stool occult blood test, rectal examination, sigmoidoscopy, pap smear, pelvic examination, breast examination, mammogram)	○	Physician	Reminder
McDonald et al. 1984, USA [30]	12467 patients	No details available	Randomized controlled trials	combined outcomes	Screening rate	Response rate for reminder	Twelve actions including 5 preventive procedures (occult blood testing, mammographic screening, weight reduction diets, influenza, and pneumococcal vaccines)	○	Physician	Reminder
Hamilton et al. 2013, UK [31]	2593 records	No details available	Nested case-control study	Lung and colorectal cancer	Screening rate and diagnosis	Diagnosis of cancer and diagnostic test	2-week referrals for lung and colorectal cancer, requested CXR, coloscopies	○	Physician	Decision support system
Murphy et al. 2015, USA [32]	733 records	60.4 ± 7.4 (intervention group)	Randomized controlled trials	Lung, colorectal and prostate cancers	Time to diagnostic evaluation	Proportion and time to diagnostic evaluation of cancer	Electronic health record-based trigger (red flag criteria of colorectal, lung and prostate cancer), Proportion and time to diagnostic evaluation of cancer	○	Physician	E-trigger
Price et al. 2019, UK [33]	No details available	No details available	Cross-sectional study	Lung and colorectal cancer	Referral rate for suspected malignancy	2-week wait referral rate	Availability and use cancer decision-support tools	×	Physician	Decision support system
Vetter, 2015, USA [34]	39 records	No details available	Non-randomized controlled trials	General diagnosis	Accuracy of diagnosis	Diagnostic accuracy and clinical documentation	Chart audit tool	○	Nurse practitioner	Decision support system
Shimizu et al. 2018, Japan [35]	100 patients	70 ± 20.9	Retrospective observational study	General diagnosis	Accuracy of diagnosis	Diagnostic error rate	Exposure to computor clinical decision support system	○	Physician	Decision support system
Burack et al. 1997, USA [36]	1225 patients	Women over 40 years old	Randomized controlled trials	Breast cancer	Screening rate	Mammography rates	Mammography rates	○	Physician	Reminder
Burack et al. 1998, USA [37]	5801 patients	Women 18–40 years old	Randomized controlled trials	Cervical cancer	Screening rate	Visitation, Pap smear	Visitation, Pap smear	×	Physician	Reminder
Sequist et al. 2009, USA [38]	21860 patients	60.3 ± 8.3 (Intervention group)	Randomized controlled trials	Colorectal cancer	Screening rate	Fecal occult blood testing, flexible sigmoidoscopy and colonoscopy	Fecal occult blood testing, flexible sigmoidoscopy, colonoscopy and detection of colorectal adenomas	○	Physician	Reminder
Litvin et al. 2016, USA [39]	No details available	Over 18 years old	Non-randomized controlled trials	Chronic kidney disease	Screening rate	CKD identification and management	Performance on chronic kidney disease clinical quality measures	○	Physician	Decision support system
Lee et al. 2007, USA [40]	1874 encounters	47.8 ± 17.88 (Intervention group)	Randomized controlled trials	Obesity	Diagnostic rate	Diagnostic accuracy of obesity-related diagnoses	Screening rate and diagnsis of obesity-related diagnoses	○	Nurse practitioner	Decision support system
Chaudhry et al. 2012, USA [41]	1763 patients	Men aged 65-75	Retrospective observational study	Abdominal aortic aneurysm	Screening rate	Screening rate of abdominal aortic aneurysm	Screening rate of abdominal aortic aneurysm	○	Physician	Decision support system
Kanealy et al. 2005, New Zealand [42]	5628 patients	Over 50 years old	Randomized controlled trials	Diabetes	Screening rate	Screening rate of diabetes	Screening rate of diabetes	○	Physician	Reminder
Wyk et al. 2008, Netherlands [43]	87886 patients	43.8 ± 14.8 (intervention group)	Randomized controlled trials	Dyslipidemia	Screening rate	Screening and treated rate of dyslipidemia	Screening and treated rate of dyslipidemia	○	Physician	E-alert
Rubenstein et al. 1995, USA [44]	557 patients	51.4 ± 18.2 (intervention group)	Randomized controlled trials	Physical function	Functional decline	Functional Status Questionnaire	Functional Status Questionnaire and completion rate of interventions	○	Physician	Feedback report
DeJesus et al. 2012, USA [45]	14674 patients	Women over 65 years old	Retrospective observational study	Osteoporosis	Screening rate	Completion rate of osteoporosis screening	Completion rate of osteoporosis screening, pratice rate of osteoporosis screening	○	Physician	Decision support system
DeSilva et al. 2020, USA [46]	13707 patients	Over 12 years old	Randomized controlled trials	Hepatitis B virus infection	Screening rate	Diagnosis of chronic HBV infection	Rate of alerts opened, test order, obtain of result, positive HBV screening test	○	Physician	E-alert
Chak et al. 2018, USA [47]	2987 patients	38.5 ± 14.7 (intervention group)	Randomized controlled trials	Hepatitis B virus infection	Screening rate	Completion rate of hepatitis B infection screening	Completion rate of hepatitis B virus infection screening, positive rate of test	○	Physician	E-alert
Sundaram et al. 2009, USA [48]	26042 patients	No details available	Randomized controlled trials	Human immunodeficiency virus infection	Screening rate	HIV screening rates	HIV screening rates, degree to guideline concordant, adherence to reminders, and provider attitude and knowledge.	×	Physician	Reminder
Miller et al. 2017, USA [49]	19869 patients	No details available	Cross-sectional study	Depression	Screening rate	Depression screening rates	Depression screening rates, contraindications to medication, level of alert, mental health risk	○	Physician	Decision support system
Downs et al. 2006, UK [50]	450 records	84.9 ± 6.6 (Intervention group)	Non-randomized controlled trials	Dementia	Rates of detection of dementia	Detection rates of dementia	Detection rates of dementia, concordance with guidelines	○	Physician	Decision support system
Ahmad et al. 2009, Canada [51]	293 patients	43.5 ± 14.8	Randomized controlled trials	Intimate partner violence and control	Screening rate	Initiation of discussion about risk for Intimate partner violence and control and detection of women at risk	Initiation of discussion about risk for Intimate partner violence and control and detection of women at risk	○	Physician	Decision support system

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
