# Peer review of "Clinical Decision Support Systems for Diagnosis in Primary Care: A Scoping Review"

_ijerph, 2021, doi:10.3390/ijerph18168435_

Round 1
Reviewer 1 Report
Thank you for the opportunity to review the manuscript ijerph-1293673.
The aim of the scoping review is to verify the usefulness of CDSS in the diagnostic domain in primary care and to identify research gaps and areas of uncertainty.
One of my little concerns related to the current review, continues to be related to the introduction literature. Greater details about previous studies results are needed than currently provided that builds the case for having conducted the current study. This could also strengthen the discussion, as it is quite common to refer to findings from those studies relative to the current review findings in the discussion and conclusions sections.
Please give a more detailed definition of CDSS, with more references.
Methods: The search strategy and the methods for data extraction, quality assessment seem appropriate.
Line 244
grammar/please improve sentence structure
Line 250-55
„The results of a retrospective analysis of the diag-
nostic process in 93 cases with a confirmed diagnosis (84 cases included collagen disease)
showed that in 53.8% of the cases, the accurate diagnosis was listed as one of the top five
differential diagnoses earlier than the physician's clinical diagnosis, and in 37.6% of the
cases, the positive diagnosis was proposed as the top of the differential, suggesting the
usefulness of the CDSS for rare diseases.“
Please make two sentences.
Line 326
Better:
In our opinion this is the first review in the literature…
Author Response
Comments and Suggestions for Authors
Thank you for the opportunity to review the manuscript ijerph-1293673.
The aim of the scoping review is to verify the usefulness of CDSS in the diagnostic domain in primary care and to identify research gaps and areas of uncertainty.
One of my little concerns related to the current review, continues to be related to the introduction literature. Greater details about previous studies results are needed than currently provided that builds the case for having conducted the current study. This could also strengthen the discussion, as it is quite common to refer to findings from those studies relative to the current review findings in the discussion and conclusions sections.
Please give a more detailed definition of CDSS, with more references.
Response:
Thank you for your constructive comments!
In response to your comments, I have added a note on CDSS in part of introduction, based on recent literature.
Methods: The search strategy and the methods for data extraction, quality assessment seem appropriate.
Line 244 grammar/please improve sentence structure
Response:
Thank you for pointing this out.
We have checked with the proofreader and revised it as “Rapid diagnosis of rare diseases by primary care physicians remains a major challenge”.
Line 250-55
„The results of a retrospective analysis of the diag-nostic process in 93 cases with a confirmed diagnosis (84 cases included collagen disease) showed that in 53.8% of the cases, the accurate diagnosis was listed as one of the top five differential diagnoses earlier than the physician's clinical diagnosis, and in 37.6% of the cases, the positive diagnosis was proposed as the top of the differential, suggesting the usefulness of the CDSS for rare diseases.“
Please make two sentences.
Response:
Thank you for pointing this out.
We have checked with the proofreader and revised it as “A retrospective analysis of the diagnostic process in 93 confirmed diagnoses (84 cases of collagen disease) showed that 53.8% of cases had accurate diagnosis as one of the top five differential diagnoses earlier than physicians’ clinical diagnosis; in 37.6% of cases, positive diagnosis was the top differential diagnosis, suggesting the usefulness of CDSS for rare diseases.”
Line 326
Better: In our opinion this is the first review in the literature…
Response:
Thank you for pointing this out.
As your suggestion, we corrected it.
Reviewer 2 Report
Dear Authors,
This is an interesting and important topic, as many health care systems globally center around the Primary Care Physician.
It would be helpful to have some information provided about the CDSS tool, as this is left unknown in the article. For example, a brief definition of the tool/s and a description of the parameters. Where are these used and how widely they are utilised globally, and how they differ? This will assist with positioning the articles retrieved and enable the reader to determine the transferability of the review findings.
Conclusions:
Can you recommend which country is leading the way in developing evidence for the effectiveness of CDSS tools and what are the key elements of a successful tool?
Author Response
This is an interesting and important topic, as many health care systems globally center around the Primary Care Physician.
It would be helpful to have some information provided about the CDSS tool, as this is left unknown in the article. For example, a brief definition of the tool/s and a description of the parameters. Where are these used and how widely they are utilised globally, and how they differ? This will assist with positioning the articles retrieved and enable the reader to determine the transferability of the review findings.
Conclusions:
Can you recommend which country is leading the way in developing evidence for the effectiveness of CDSS tools and what are the key elements of a successful tool?
Response:
Thank you for your constructive comments.
Information about the CDSS and its worldwide use, which country is leading the way in developing evidence, key for the effectiveness of CDSS has been added to introduction based on recent literature.
Reviewer 3 Report
Thank you for the opportunity to review this manuscript. The study topic about the applicability of clinical decision support systems to promote patients safety is a very relevant topic.
This manuscript provides relevant information about CDSS However, the article needs to be worked on. I would like to make a few suggestions for revision”
This article aims to provide an overview regarding the usefulness of “CDSS in the diagnosis domain in primary care and identify research gaps…”
- The introduction was based in works published between 2008 and 2017. Please can you update the introduction with more recent works.
“The search query returned 580 articles and backward citation tracking included 22 87 articles. After analyzing the titles and abstracts, 551 articles that were not relevant to the 88 research question were discarded. Finally, 51 full-text articles were reviewed, among 89 which 25 articles that did not satisfy the eligibility criteria were excluded”
- The search query did not retrieve any duplicated work?
- The flow diagram should include the number (551) of articles discharged.
- Please can you provide a table or adapt table one with information regarding the number of participants included in each study, the mean age of the patients, the main outcomes measured, the main observation of the articles, and the impact associated with CDSS.
- Please can you provide information regarding the quality of the included studies.
Author Response
Thank you for the opportunity to review this manuscript. The study topic about the applicability of clinical decision support systems to promote patients safety is a very relevant topic.
This manuscript provides relevant information about CDSS However, the article needs to be worked on. I would like to make a few suggestions for revision”
This article aims to provide an overview regarding the usefulness of “CDSS in the diagnosis domain in primary care and identify research gaps…”
1. The introduction was based in works published between 2008 and 2017. Please can you update the introduction with more recent works.
Response:
Thank you for your helpful suggestion. We agree with your comment.
Thank you for your comments. I have added a note on CDSS in part of introduction, based on recent literature.
“The search query returned 580 articles and backward citation tracking included 22 87 articles. After analyzing the titles and abstracts, 551 articles that were not relevant to the 88 research question were discarded. Finally, 51 full-text articles were reviewed, among 89 which 25 articles that did not satisfy the eligibility criteria were excluded”
2.The search query did not retrieve any duplicated work?
Response:
Thank you for pointing that out.
To avoid duplicated work, multiple authors conducted searches, manually reviewed for duplicates, and removed duplicates where necessary.
3.The flow diagram should include the number (551) of articles discharged.
Response:
Thank you for pointing that out.
As your suggestion, we corrected it.
4.Please can you provide a table or adapt table one with information regarding the number of participants included in each study, the mean age of the patients, the main outcomes measured, the main observation of the articles, and the impact associated with CDSS.
Response:
Thank you for your constructive comments.
In response to your suggestion, we have added information on the number of participants, their ages, the main outcomes measured, and the main observations of the paper.
Impact associated with CDSS is a very interesting factor. Meanwhile, it is highly heterogeneous because the amount of discussion on impact associated with CDSS varies considerably from study to study and the definition of CDSS is set quite broadly. Therefore, we did not add it to the table because we thought that mentioning it in the table would make it difficult for readers to understand.
5.Please can you provide information regarding the quality of the included studies.
Response:
Thank you for your constructive comments. In this scoping review, we have not attempted a detailed quality assessment in accordance with PRISMA-ScR. ( http://www.prisma-statement.org/Extensions/ScopingReviews ) .Therefore, we would not include it in the content of this scoping review.